# Impact of Plant-Beneficial Bacterial Inocula on the Resident Bacteriome: Current Knowledge and Future Perspectives

**DOI:** 10.3390/microorganisms10122462

**Published:** 2022-12-13

**Authors:** Francesco Vuolo, Giorgia Novello, Elisa Bona, Susanna Gorrasi, Elisa Gamalero

**Affiliations:** 1Sacco s.r.l. via Alessandro Manzoni, 29a, 22071 Cadorago, Italy; 2Dipartimento di Scienze e Innovazione Tecnologica, Università del Piemonte Orientale, 15121 Alessandria, Italy; 3Dipartimento per lo Sviluppo Sostenibile e la Transizione Ecologica, Università del Piemonte Orientale, 13100 Vercelli, Italy; 4Center on Autoimmune and Allergic Diseases (CAAD), Università del Piemonte Orientale, 28100 Novara, Italy; 5Dipartimento di Scienze Ecologiche e Biologiche, Università degli Studi della Tuscia, 01100 Viterbo, Italy

**Keywords:** PGPB, resident microbiota, biofertilizer regulation, *Azotobacter*, *Azospirillum*, *Rhizobia*, *Pseudomonas*, *Bacillus*, bacterial consortia

## Abstract

The inoculation of plant growth-promoting bacteria (PGPB) as biofertilizers is one of the most efficient and sustainable strategies of rhizosphere manipulation leading to increased plant biomass and yield and improved plant health, as well as the ameliorated nutritional value of fruits and edible seeds. During the last decades, exciting, but heterogeneous, results have been obtained growing PGPB inoculated plants under controlled, stressful, and open field conditions. On the other hand, the possible impact of the PGPB deliberate release on the resident microbiota has been less explored and the little available information is contradictory. This review aims at filling this gap: after a brief description of the main mechanisms used by PGPB, we focus our attention on the process of PGPB selection and formulation and we provide some information on the EU regulation for microbial inocula. Then, the concept of PGPB inocula as a tool for rhizosphere engineering is introduced and the possible impact of bacterial inoculant on native bacterial communities is discussed, focusing on those bacterial species that are included in the EU regulation and on other promising bacterial species that are not yet included in the EU regulation.

## 1. Introduction

The term holobiont, introduced by Lynn Margulis in 1991 [1] who proposed the endosymbiotic theory, indicates “*a simple biological entity involving a host and a single inherited symbiont*”. This concept was then enlarged to include the host and its associated microbial community, thus highlighting the occurrence of microorganisms in almost every kind of environment and their fundamental role related to the fitness and function of the host and to its evolution, as well as their possible co-evolution [2]. In this context, the genome of the host, together with that of the members of the associated microbial communities, has been defined as the hologenome [3]. It is universally accepted that plants, through root exudation, create a nutritional hotspot, identified as the soil surrounding the roots (rhizosphere) where specific bacterial populations are recruited and, in turn, establish beneficial, neutral, or deleterious relations with the host plants [4]. This concept is widely known as the rhizosphere feedback loop and it its realization is one of the main variables at the base of the different bacterial communities found in bulk soil and in the rhizosphere [4,5,6]. The degree of intimacy that these bacteria can establish with the plant can vary from epiphytic microorganisms, living on the root/leaf surface, to endophytic ones, colonizing the internal plant tissue without inducing any negative effect on plant health and development [7,8]. The microbial communities associated with the plant may be viewed as a collection of microorganisms sharing the same habitat. However, this definition neglects the occurrence of intricate relationships established among the different members of the microbial community. Therefore, the term microbial community currently refers to “*multi-species assemblages, in which (micro) organisms interact with each other in a contiguous environment*” [9].

The set involving the plant-associated microbial communities and the interactions occurring among them is indicated as the microbiota, where the host, together with its microbiota, constitute the previously defined holobiont. For a long time, the words microbiota (the pool of microbial taxa associated with a host, including bacteria, archaea, lower and higher eukaryotes, and viruses) and microbiome (the collections of the genes of the members of the microbiota) have been used as synonyms and this has generated some confusion about the use of the terminology. To avoid any further uncertainty about the definition of these terms, in 2015, Marchesi and Ravel published an editorial focused on the vocabulary of microbiome research [10].

Currently, the availability of whole-genome sequencing and -omics techniques such as metagenomics, metatranscriptomics, and metaproteomics led to a real revolution in the field of microbial ecology [11], and allows one to obtain information about the functioning and behavior at the single strain level as well as at the community level. Molecular studies based on 16S rDNA sequencing estimated that the bacterial density in one gram of soil is 10^9^ cells and that 4  ×  10^6^ different microbial taxa occur in each ton of soil, with the fraction of the bacteria able to grow and develop colonies on synthetic media estimated to be ~1–10% of the total [12,13,14]. However, traditional microbiological methods based on culture have not been abandoned and the so-called culturomics, based on the study of culturable bacteria, received a renewed interest by the scientific community. In fact, the culturable fraction of soil microorganisms represents a precious source of bacterial strains that can be exploited in applied environmental microbiology as biofertilizers, biocontrol, and bioremediation agents [15].

The use of plant-beneficial microorganisms as biofertilizers is becoming more and more important, especially considering the effects of global warming on cultivated lands. It is estimated that the Earth’s temperature has risen by 0.08 °C every 10 years starting from 1880. Moreover, predictive models indicate that by 2100, the mean temperature of the planet will be 1.1–5.4 °C warmer than today (https://www.climate.gov, accessed on 11 December 2022). As a direct consequence, global drought has recently increased dramatically (https://climate.nasa.gov/news/3117/drought-makesits-home-on-the-range/, accessed on 5 January 2022; https://www.c2es.org/content/drought-and-climate-change/, accessed on 5 January 2022), especially in the Eastern Mediterranean and Middle East zones, where an increase of 3.5–7 °C is expected by the end of the century [16,17]. The scarce availability of water (i.e., drought) in many geographic zones, exacerbated by the irrigation of lands, induces the accumulation of salt in soil. Drought and salinization are currently the main environmental stresses impairing crop productivity. On the other hand, the world population is currently growing by approximately 83 million people annually, thus enlarging the gap between global food production and food demand. In addition, the distribution of the food resources at the global level is strongly unbalanced (https://www.fao.org/fileadmin/templates/wsfs/docs/expert_paper/How_to_Feed_the_World_in_2050.pdf, accessed on 11 December 2022). In such a context, several commercial biofertilizer formulations, mainly represented by *Rhizobium* spp., *Azotobacter* spp., and *Azospirillum* spp., have been produced. The global biofertilizer market constitutes only a very small fraction of the synthetic agrochemical market; the global biofertilizers market has been valued at USD 2.6 billion in 2021 and it is expected to reach USD 3.5 billion by 2025 [18].

As a result of the increasing interest in biofertilizers, the literature abounds with papers reporting the positive effects of microbial inoculants on plant health, biomass, production, and the increased nutritional value of edible seeds and fruits, both under controlled and open field conditions and under different soil management regimens [19,20,21,22]. The enthusiasm of the scientific community regarding these results has frequently overshadowed the concerns about the impact of plant inoculation on the soil/rhizosphere microbial community. Consequently, the impact of the deliberate release of biofertilizers on the resident bacterial community is rarely monitored and information about this approach is relatively rare and often contradictory. The aim of this review is to fill this gap, focusing on the impact of beneficial bacterial inoculants, alone or combined in artificially constructed consortia, on the native microbiota.

## 2. Plant Growth-Promoting Bacteria: A Brief Description of the Main Mechanisms by Which They Favor Plant Development and Improve Plant Health

Plants are surrounded by a highly complex bacterial diversity, which is composed of different genera and species that can interact with the host plant in a neutral, harmful, or beneficial way. The beneficial bacteria are referred to as plant growth-promoting bacteria (PGPB) and include rhizospheric, epiphytic, and endophytic microorganisms living on/in the roots, shoots, leaves, flowers, and fruits. Members of various bacterial genera have been recognized for their plant growth-promoting features, including *Bacillus, Paenibacillus, Pseudomonas, Rhizobium, Enterobacter, Serratia, Agrobacterium, Azospirillum, Azotobacter, Klebsiella, Burkholderia, Arthrobacter, Micrococcus, Microbacterium*, and *Rhodococcus* [23]. PGPB modulate the overall plant fitness by increasing the plant biomass, enhancing nutrient availability, providing protection from phytopathogens, and promoting the tolerance to abiotic stresses (e.g., soil acidity/alkalinity, soil salinity, drought, flooding, low/high temperature, the presence of inorganic and organic pollutants in the soil, low nutrient bioavailability, and excessive radiation) [24,25,26,27]. In general, the mechanisms of plant growth promotion are categorized into direct and/or indirect (Figure 1). Direct mechanisms include the production of phytohormones and activities that increase the availability and uptake of nutrients, whereas indirect mechanisms consist of activities conferring protection against pathogens and tolerance to various abiotic stresses [8]. PGPB may use either direct or indirect mechanisms (or both), one or more mechanisms at the same time, as well as different mechanisms at different stages of host plant development [25]. A brief overview of some of the main mechanisms is outlined below and explained in Figure 1.

About 16 micro- and macro-nutrients are fundamental for plant development, and a lack of any of these nutrients might result in an imbalanced growth and consequent plant yield reduction [28]. Nitrogen fixation, phosphorus solubilization, and siderophores production are recognized as the major beneficial mechanisms involved in nutrient provision to the plant.

Nitrogen is one of the major nutrients required for plant growth, being an essential constituent of proteins, nucleic acids, and many other biomolecules. Nitrogen deficiency can result in a disturbed root–shoot ratio, short lateral branches, small leaves, the acceleration of leaf senescence, and decreased photosynthesis, which heavily affect plant fitness and crop yield [29]. Although there is a large nitrogen reservoir in the Earth’s atmosphere, the atmospheric nitrogen cannot be directly used by plants and must be reduced to ammonia. Its bioavailability principally relies on biological fixation, and the conversion into ammonia is carried out by nitrogen-fixing prokaryotes via the nitrogenase enzyme complex [30]. The PGPB involved in biological nitrogen fixation can be either symbiotic or non-symbiotic bacteria. Among the symbiotic bacteria, rhizobia (e.g., *Rhizobium, Sinorhizobium, Bradyrhizobium*, and *Mesorhizobium*) establish associations with leguminous plants and the Actinomycetes *Frankia* establish a symbiotic association called actinorrhizae with a number of dicotyledonous plants [31,32]. Finally, non-symbiotic bacteria may be either free-living or endophytic and include members of *Azospirillum, Gluconacetobacter, Azoarcus, Herbaspirillum, Azotobacter, Pseudomonas, Enterobacter,* and cyanobacterial genera (*Nostoc* and *Anabaena*) [25,33].

Due to its low bioavailability, phosphorus is the second most important nutrient limiting plant growth. It represents a key component of many biological molecules and is essential for plant development. Depending on the plant species, phosphorus deficiency can have various deleterious effects, such as stunted plant growth, decreased leaf expansion, upward tilting and curling of leaves, anthocyanin accumulation, and delayed plant maturity [34]. Despite its high abundance in the soil, in both inorganic and inorganic forms, most of the phosphorus is not available for the plant, occurring in insoluble forms. Moreover, much of the soluble inorganic phosphorus provided by chemical fertilization is immobilized soon after the treatment, becoming unavailable to plants [35]. Phosphate-solubilizing bacteria may help to overcome these issues, providing bioavailable forms to the plants through the solubilization of inorganic phosphorus (via low molecular weight organic acids) and/or the mineralization of organic phosphorus (via phosphatases catalyzing the phosphoric ester cleavage) [36]. In addition, some strains can exhibit both phosphate-solubilizing and -mineralizing features [37,38].

Being a component of proteins involved in vital metabolic processes such as photosynthesis, respiration, and nitrogen fixation, iron is a nutrient required by all living organisms. Notably, its deficiency is one of the most important factors affecting crop production in the world. Iron starvation leads to chlorosis, decreased photosynthesis, and reduced plant yield and nutritional quality [39]. Although iron is one of the most abundant elements on the Earth’s surface, it is largely present as insoluble forms, which cannot be assimilated by plants and soil microorganisms in sufficient amounts to support their growth [25]. To overcome this problem, some bacteria produce siderophores, low molecular weight chelating molecules that allow to increase iron solubility. The soluble ferric iron–siderophore complex can easily enter the bacterial or plant cells through specific receptors; iron will be then released from the siderophore either by reduction to the ferrous state or through siderophore cleavage [40,41].

Major direct mechanisms include the production and modulation of phytohormones (e.g., auxin, cytokinin, gibberellin, abscisic acid, ethylene, salicylic acid, and jasmonic acid). Many PGPB are able to synthesize and/or modulate the level of some of these molecules, supplying an exogenous phytohormone supplementation that contributes to promoting and/or influencing the growth, development, and differentiation of the plants [42,43,44]. Among the phytohormones of bacterial origin, most of the attention has focused on the role of auxin, in particular indole-3-acetic acid (IAA). It has been suggested that ~80% of rhizosphere bacteria can produce IAA [45]. IAA is principally a molecular trigger of the division, extension, and differentiation of plant cells and tissues. In addition, it controls various stages of plant development (e.g., seed and tuber germination, lateral and adventitious root formation, and vegetative growth processes), as well as plant geotropic and phototropic responses [46]. Among them, the enhanced formation of lateral and adventitious roots is one of the main beneficial effects of the bacterial IAA, which leads to an improvement in nutrient uptake and root exudation and then increasing bacterial proliferation around the roots [47,48]. In addition, the IAA produced by nodulation-enhancing rhizobacteria and various rhizobia strains enhance nodulation in leguminous plants and improve nitrogen fixation [49].

Plants experience various abiotic (e.g., drought, salinity, alkalinity, and inorganic and organic contaminants) and biotic (pathogenic bacteria, fungi, viruses, nematodes, and insects) stresses, which can severely impact their fitness and health. PGPR contribute to the overall health status of the plants through a series of mechanisms that reduce or prevent the deleterious effects caused by these stresses. Under stressful conditions, the endogenous levels of plant ethylene notably increase (referred to as “stress ethylene”) with consequent inhibitory effects on the overall plant growth [50]. This phytohormone is a key regulator of various plant growth processes (e.g., fruit ripening, flower senescence, and leaf and petal abscission), but it is also involved in plant defense responses to stresses [51]. Nevertheless, it has been shown that the modulation of the 1-aminocyclopropane-1-carboxylic acid (ACC; precursor of ethylene) levels by the ACC deaminase of bacterial origin reduces the ethylene amount, mitigating the detrimental effects of environmental stresses such as drought, salinity, flooding, heavy metals, and pathogens [46,52]. In addition to ACC deaminase synthesis, PGPB use other mechanisms to improve plant tolerance to abiotic stresses: bacterial phytohormones are involved in plant responses to various stresses, including drought, salinity, extreme temperature, and heavy metals [43,53], while bacterial siderophores may help to alleviate abiotic stresses, such as high levels of heavy metals in soil [54].

PGPR also express various biological control traits that can prevent harmful effects of phytopathogens and indirectly promote plant growth. These activities involve competition for nutrients and niche exclusion, antibiosis, lytic enzyme production (e.g., chitinases, β-1,3 glucanases, cellulases, and proteases), and induced systemic resistance (ISR) [55]. For instance, siderophore-producing rhizobacteria can prevent soilborne phytopathogen proliferation around the root. These bacteria compete for iron in the soil and reduce its availability through siderophore action, negatively impacting the growth of other microorganisms [56]. Several PGPB have been reported to produce various metabolites (e.g., cell wall lytic enzymes, antibiotics, and hydrogen cyanide) effective in plant protection against phytopathogens, resulting in disease development prevention [57,58]. Beneficial bacteria can also act as inducers of ISR: their presence significantly induces resistance, thus enhancing plant protection and leading to a reduced disease incidence [59].

### 2.1. PGPB Selection and Formulation Process

PGPB have often variegated origins and are actually isolated from a wide variety of sources, which not only belong to the plant tissue, rhizosphere, or bulk soil. Many PGPB are indeed found as commensals or inactive hubs in some multi-microbial complex networks on food surfaces, wastewaters, sludge, and mixed environmental samples. Nevertheless, although these microbes could potentially turn out to be functional ingredients for a biostimulant product (mainly due to their flexibility to adapt to a fluctuating environment), isolating autochthonous microbial species from plant tissue or rhizosphere represents an ideal condition to guarantee a natural adaptation of the working strain to a specific crop or precise environment.

Once isolated and uniquely taxonomically assigned, the candidate strains need to be tested for the physiological features of interest, which can range from multiple “biostimulatory activities” to preselected biocontrol against targeted pathogens. In both cases, the first level of analyses starts in vitro, with several plate-assay methodologies, which could, in some cases, be made high-throughput, to run multiple parallel trials (considering the initial high numbers of potential candidates extracted from mixed sources). This pipeline has many limitations, the first of which is the reduced percentage of microbes that can be propagated on culture media in vitro compared to the total microflora. In fact, when trying to identify and propagate microbes in vitro, a significant selection of the whole microbial population, likely representing only a fraction of the entire diversity, is applied [60,61]. In fact, only 0.01–10% of the soil microbiota is culturable, while the remaining portion is represented by live and active cells classified as viable, but not culturable (VBNC) [62,63]. To mitigate this flaw, different approaches had been pursued: (a) plating and isolating the microbes from different cultivation media, and (b) developing new culture media compatible with as many microbial families as possible. In the view of the latter attempt, de Raad et al. [64] very recently developed a newly defined medium that supports the growth of 108 diverse bacterial species belonging to 36 different families.

Moreover, their medium is also compatible with exometabolomics profiling, which is very helpful to understand the mechanistic bases of the plant–microbe and microbe–microbe interactions. Once isolated as a single strain, this is tested for several PGP activities, such as: siderophore production; indole-3-acetic acid synthesis; nitrogen fixation; phosphate solubilization; ACC deaminase activity; β-glucanase assay; HCN synthesis; ammonia production; and biocontrol activity against targeted pathogens [14,30].

Having characterized the microbial isolates for their biostimulatory/bioprotection activity, their efficacy will be tested in vitro, and finally in vivo (preferably in field trials), to ultimately prove their efficacy. However, before planning on applying the microbial inoculants, it is recommended to consider important safety and health features of the strain of interest, such as the possible production of toxins, antibiotics, and resistance genes likely to be transferred (see next section) [61,65,66].

Additionally, another key step before moving the trials from in vitro to in vivo is to study and apply an effective formulation for the microbial inoculant, which must address the following targets: (a) to maximize the activity of the microbe at the phyllosphere/rhizosphere/rhizoplane/seed surface; (b) to guarantee its interaction with the plant tissue; (c) to protect the microbe against possible biotic/abiotic stress factors (e.g., viruses, UV light, heat/cold); and (d) to extend the shelf life of the final product [61,67,68]. In reality, the formulation has to deal with many additional challenges and limitations, mainly connected with the availability of the raw materials, the regulatory constraints, and the common applicative practice linked to the target crops. For example, foliar fertilizers for maize will differ from the ones for lettuce or soybean, due to a different leaf surface biochemistry (which might require higher adhesion, for example) [68,69]. Nevertheless, there are common guidelines to drive the selection of the different ingredients in a formulation such as, for example, the use of sugars and polysaccharides to increase the viability and vitality of the microbes within the product and in the soil, respectively [70,71]. Similarly, trehalose or betaine can be used as a mitigation agent to prevent desiccation [72,73,74].

Additionally, protein hydrolysates or humic acids are used to both extend spore viability and nourish the soil [75,76,77]. In conclusion, the formulation represents the conceptual boundary between basic and applied microbial ecology, and it is a major factor driving the efficacy of the microbial inoculant and the final product itself. Thus, it would be beneficial to implement the current microbiological knowledge in the formulation to design an ecologically sustainable and knowledge-based solution with maximum efficacy.

#### EU Regulation for Microbial Inoculants

As important as the scientific rationale behind the product development is, it is the definition of the regulatory boundaries in which to move for a correct fertilizer production process. Since the entry into force of the EC 2003/2003 law, there was a single European regulatory guideline for fertilizers, exclusively covering products derived from inorganic, mined, or chemical origins. On top of this law, each of the EU countries developed national laws to regulate all of the left-out product categories. This mixed legislative scenario generated many gray areas and ambiguities, and great diversities on how the products were defined, which specifications were set, and on the contaminants limit. The technological advancements of the last 20 years had also been radical, and so the individual national laws needed constant updates and addenda.

In an attempt to harmonize and unify, the EC designed the 2019/1009 law, the aims of which are threefold: (a) to adopt a single and unambiguous standard for all of the current fertilizers on the market; (b) to promote the circular economy principles, enacted by the EU in 2015; and (c) to foster the continental fertilizer markets, with common and undersigned principles to safeguard the ecological balance of the soil environment. In this way, safety, quality and labeling are now referred to a single shared document by all European fertilizer producers.

The EC 2019/1009 law has set the new standards through seven product function categories (PFCs): PFC1 for all fertilizers (inorganic, organic, and organo-mineral ones); PFC2 for liming materials; PFC3 for soil improvers; PFC4 for growing medium; PFC5 for inhibitors; PFC6 for plant biostimulants; and PFC7 for fertilizing product blends. The PFCs were designed with the clear intent of including all kinds of currently existing products, and at the same time allowing areas for the creation of new fertilizers that could be freely commercialized by EU members. At the same time, each producer is not forced to CE-label their products, as the national laws remain active. However, having a product that only complies with national law would require a regulatory check for mutual recognition compatibility each time the manufacturer of that product would like to commercialize in another EU country. Thus, the EC 2019/1009 law sets the base for a more direct scheme of regulation/commercialization.

Beyond the pros previously described, the current law has been carefully designed to limit the excessive use of the products themselves to reduce leaching, increase nutrient efficiency, and cut wastes. Moreover, the quality standards are tightly controlled to stop the eutrophication and contamination caused by the over-intensive use of fertilizers. Finally, to reduce extensive mining, to find alternative sources of nutrients, and to match green-deal criteria, EC 2019/1009 promotes circular economies, which are a key solution to the current shortage of fertilizers and raw materials, and a potential booster for SMEs and a support for agriculture, especially in the light of climatic changes.

Despite the many positive points, several fertilizer producers and also many scientists noticed margins of improvement for EC 2019/1009. For example, the PFC6 regulates biostimulants, in which the microbial inoculants are a novel and leading trend in the market. Nevertheless, the law currently allows only four microbial types, three of which are bacteria belonging to the genera *Azospirillum*; *Azotobacter* and *Rhizobium*; and one represented by arbuscular mycorrhizae. This is a quite restrictive selection, as both literature and commercially available products showed how many more PGPs belong to different families compared to the ones included in the PFC6. Moreover, one of the main goals of the present law was to find alternative sources of P and nutrients, to reduce natural resource exploitation. Unfortunately, the bacterial inoculants of the PFC6 express little holistic biostimulation, while being beneficial mainly through nitrogen fixation and nutrition [78,79,80]. To stimulate the market and offer more complete, variable, and effective products, it would be beneficial to extend the list of beneficial microbes to more species, as long as they respect the safety standards.

### 2.2. PGPB Inoculant as a Tool for Rhizosphere Engineering

As stated in the first section of this review, the rate of food production is increasing slower than the food demand, especially in certain geographical areas. The problem is not only limited to the amount of available food, but also to the quality of food (nutrient value). It is widely accepted that there is a link between the nutrients introduced with the diet and human wellbeing. At the same time, there is a tight relationship between the human and the plant microbiome. While bacterial members of the plant and human microbiomes frequently overlap each other, it is also true that microorganisms resident in/on fruits and vegetables reach and join the gut microbiome, thus affecting its composition with relevant consequences on human health [81]. Similarly, to what happens during fecal transplant in humans, using beneficial microorganisms or consortia of microorganisms for plant inoculation leads to improved plant growth, yield, and biomass development [81]. Therefore, the manipulation of the plant microbiome through the use of PGPB and the exploitation of their physiological traits (described in Section 2) is one of the possible strategies to engineer the rhizosphere and reduce the application of chemical inputs in agriculture. However, according to a quite old, but always actual view, the rhizosphere is recognized as a tripartite entity (the so-called “rhizosphere trinity”) composed of the soil, the plant, and the associated microbiota and the relations occurring among these three components (Figure 2) [82]. Obviously, the rhizosphere environment can be different according to the plant species, phenological stage, and health status; similarly, soil with different fertility, management, and chemical/physical parameters affects plant growth as well as the development of the plant-associated microbiome (Figure 2). In turn, the soil and rhizosphere microbiome can show evident effects on the plant and on the soil composition. As a direct consequence, the rhizosphere can be engineered by manipulating the microbiota, the plant (through the selection of cultivar more resistant to environmental stresses or containing different amount of useful nutrients, plant breeding, and genetic modifications) and modifying soil parameters (by soil amendments with compost, manure, biochar, or sewage sludge) (Figure 2). Although these three compartments are obviously interconnected, in this review, we will focus our attention on the manipulation of the microbiota.

There are two main strategies to engineer the microbiota: (i) manipulate the resident microorganisms or (ii) introduce new bacterial strains into the native microbiota [83]. The first strategy is based on the idea that the use of one or more bacterial strains (single-strain inocula or consortia) isolated from and well adapted to a specific plant species and/or environmental conditions can lead to increased rhizosphere competence and survival in open fields, especially under stressful conditions [84]. In this way, the introduction of allochthonous or foreign species behaving as invaders is avoided as well as the consequences at ecosystem level, thanks to the capability of the plant to attract beneficial microorganisms.

The second approach is based on the bacterial strain selection according to the capability to efficiently colonize the plant that are subsequently used as plant inocula in different plant organs during different phenological stages and in diverse environmental conditions [85]. In this context, the improvement of the bacterial strain’s capability and the expression of plant-beneficial physiological features can be improved by genetic manipulation through a bottom-up or top-down flowchart [86]. Following the bottom-up strategy, microorganisms are isolated from plant organs, genetically transformed in order to express the desired trait, and pooled together in a synthetic community that is then used as an inoculum. The top-down strategy is based on the in situ incorporation of desired genetic traits in a wide range of hosts via horizontal gene transfer [86].

The main advantage of building up and using a synthetic community is the possibility to exploit diverse physiological futures expressed by the members of the consortia, and then compensating for the possible deficiencies of individual bacterial strains. However, formulating an efficient consortium represents a relevant challenge, especially considering that in a very simple consortium composed of two bacterial strains, at least six types of relationships such as commensalism, competition, predation, neutralism, cooperation, and amensalism can be established [87,88]. Moreover, the complexity of the possible interactions among the members of a consortium become higher as the number of members increases. By considering a consortium composed of three or four bacterial strains, it has been estimated that about 729 and 531.441 interactions, respectively, can occur [87,88]. Obviously, in order to realize an efficient consortium, positive relationships must be maximized, while negative interactions should be minimized and this is a very relevant challenge. Moreover, the formulation of an efficient synthetic community goes through i) the definition of the bacterial density to be applied to seeds, seedlings, and plants, ii) the selection of the best carrier, formulations, and storing conditions, iii) the identification of the best time of inoculation, iv) the assessment of the plant colonization (from the recognition, to the attachment phase, to the possible internal spreading), v) the evaluation of the performance and survival under different soil types/managements or environmental stressful conditions, and vi) the final effects on plant biomass and yield.

## 3. Impact of Bacterial Inoculant on Native Bacterial Communities

The release of bacterial strains in the environment is currently a quite common practice; it is used to clean up polluted soil, suppress soil borne diseases, promote plant growth, and restore biodiversity. As previously stated, the PGPB inoculation of plants exploits their beneficial effect on plant development and on ecosystem functions; this represents a sustainable solution for the reduced plant yield related to climate change [89]. However, once inoculated on a seed or on a plant, the PGPB, behaving as an invader, can potentially induce shifts in microbial communities, thus perturbing the niche previously created by the resident microbiota [90]. Once microbial invasion occurs, three main outcomes can be expected: (i) the invader can stably establish within the native microflora and then induce shifts in the microbial community composition, (ii) the resilience of the soil leads to the elimination of the invader and restores the initial condition, thus maintaining the community as it was before the invader’s arrival, and (iii) the invader can establish in the native microflora and then induce transient shifts in the microbial community composition followed by the restoration of the initial conditions (Figure 3) [91,92,93].

In the case where the invader creates a stable interaction with the resident bacterial community, it can develop positive or negative relationships with the members of the community, thus leading to changes in species composition. These modifications are not only restricted to the community level; cascading effects can spread to the ecosystem level and then induce unpredictable and perhaps undesired consequences on the agroecosystem functioning [89]. Therefore, plant inoculation with PGPB can lead to the so-called “legacy effects” [94]. This term was introduced in the 1990s [95,96] and is generally used to indicate the “impacts of a species on abiotic or biotic features of ecosystems that persist for a long time after the species has been extirpated or ceased activity and which have an effect on other species” [97]. According to Liu et al. [89], there are three ways by which plant-beneficial bacterial inoculants can induce a legacy effect on agroecosystems in the context of climate change. The first one consists of the direct effects of inoculants showing specific functional traits (e.g., the release of exopolysaccharides increasing water holding capability) that are relevant to climate change [36,51,89]. The second one is based on those indirect effects realized by the PGPB affecting plant growth and development (e.g., the synthesis of ACC deaminase and auxins). The third pathway is based on the alterations induced by the plant-beneficial bacterial strain on the soil microbial community. This can occur through the modification of the root exudate composition and/or allocation or establishing diverse relationships (competition, antagonism, or mutualism) with the members of native communities, thus modifying the relative abundance of less prevalent taxa [98]. Once the bacterial inoculant establishes in the microbiota, the density of the whole microbial community can increase, at least to the level of the inoculated taxon [99]. If the introduced plant-beneficial microorganism outcompetes the members of the resident community, the biodiversity decreases and the inoculant becomes dominant [100]. Alternatively, plant inoculation with PGPB can lead to an increase of biodiversity if the dominant taxa are outcompeted by the PGPB itself [90]. When the PGPB improves the plant nutrient status, it can favor a density increase of the resident taxa. However, one of the main problems associated with the use of PGPB as plant inoculants is their brief persistence in the soil or rhizosphere, a factor that frequently causes low efficiency as biofertilizers or biocontrol agents and inconsistent results, especially in open field conditions. It can therefore be expected that plant inoculation with PGPB results in transient microbial load leading to an impact on the native community that weakens over time (Figure 3). However, the rapid disappearance of inoculated PGPB in the soil or rhizosphere does not necessarily correspond to the lack of a lasting legacy effect on the native microflora [92,93,94,95,96,97,98,99,100,101]. Despite the relevance of this topic, in analyzing the literature available in the main scientific databases by matching the terms biofertilizers/PGPR/legacy effects, we found only four published manuscripts.

A June 2019 paper reporting a meta-analysis of the existing literature about the impacts of PGPB inoculants on the native microbial community highlighted 445 papers in the Web of Science database [99]. Currently (August 2022), in the Web of Science database, there are 728 papers, demonstrating the increasing interest on the possible non-target effects of plant inoculants on resident microorganisms. Based on the data presented by Cornell et al. [99], the soil bacterial community appears to be more prone than the fungal community to change after the inoculation of plants with microbes. This information suggests that the fungal component is more stable and less sensitive than the bacterial fraction to environmental perturbations induced by the inoculation of plants with PGPB. Moreover, the number of papers reporting increased bacterial biodiversity following plant treatment with beneficial microorganisms was higher than that showing a reduction of biodiversity in the bacteriome. Although the variations of the bacterial biodiversity were not related to the type of inoculant (i.e., bacterial vs. fungal), microbial consortia appear to be more likely to induce alterations of the bacterial biodiversity than an inoculant composed of a single species. Thus, the ways in which the bacteriome can be perturbed following plant inoculation with a PGPB are different and complex, and can vary according to different factors such as the host plant species, the availability of nutrients in the soil, the type and composition of the inoculant, the bacterial species included in the inoculant, the commercial formulation used, and the plant’s response to the inoculant. Given the high complexity level, in the following sections, we decided to focus on the possible effects imposed by *Azospirillum* spp., *Azotobacter* spp., and *Rhizobium* spp. on the native bacterial community. These three bacterial genera are of great interest given that their use as biofertilizers in open field conditions is allowed by European regulations (EU regulation 2019/1009). Therefore, several examples regarding the effects of these PGPB on the soil bacteriome will be given. Moreover, it is important to consider the impact expressed on the resident bacterial community by other genera, such as *Bacillus* and *Pseudomonas*, which are well known as PGPB and biocontrol agents and deserve more attention for their role as biofertilizers and their possible future acceptance by European regulations. Finally, information on the impact of bacterial consortia on the native microbial community are provided in the last section of this manuscript.

### 3.1. Effects of Azospirillum Inoculation on Resident Microbiota

Bacteria belonging to the genus *Azospirillum* are free-living PGPB that affect the growth and yield of several plants, including species of agronomic importance [102]. *Azospirillum* is a genus of diazotrophic bacteria, which often show the capability to solubilize phosphate and produce phytohormones, ACC deaminase, and siderophores [27,103]. The promoting activity of both the single and combined inoculant of *Azospirillum* species/strains is well known; at the same time, several studies focused on the inoculant effects on resident microbiota, showing variable results according to the susceptibility and buffering ability of the indigenous microbial communities. Some investigations reported that the inoculation resulted in minor or nearly undetectable changes in root and rhizosphere microbial communities, whereas others demonstrate notable shifts in resident microbiota. As can be deduced by the study outcomes reported hereafter and as stated by various authors, these observed variable effects can be due to the differences in experimental conditions, including levels of the introduced inoculant, inoculant species/strains, host plant species/genotypes, plant age, and soil type.

Various work focused on the impact of some *Azospirillum* spp. inoculants on maize rhizosphere microbiota. A series of studies carried out under greenhouse conditions, where maize plants grown in two different soils were treated using two different inoculants (*A. brasilense* Cd; *A. baldaniorum* Sp245, formerly *A. brasilense* Sp245), highlighted a very marginal effect on rhizoplane and rhizosphere bacterial populations upon inoculation [104,105,106]. The authors investigated the possible impact on the whole bacterial community [104,105,106] as well as on specific groups (*α-Proteobacteria*, *Actinobacteria*, *Bacteroidetes*, *Pseudomonas*, and *Bdellovibrio*) known for their significant roles in nutrient cycling in the rhizosphere [105], using DGGE and ARISA fingerprinting methods. Both abundant bacteria and specific bacterial populations appeared weakly affected by inoculation, whereas plant age seemed to strongly influence the native rhizobacteria.

Conversely, in other studies, different responses from transient, but significant changes to strong shifts in rhizosphere microbial communities of field-grown maize have been observed. Pandey et al. [107] found that inoculation with two *A. brasilense* strains (SP7 or GYNL), even if not affecting the size of the total bacteria and fungi, stimulated certain beneficial rhizobacterial populations. An approximately 2- to 4-fold increase in *Actinomycetes* and 1.5- to 3-fold increase in free-living nitrogen-fixing bacteria was recorded in the middle of the maize growing period (at 40 and 80 days from sowing).

Di Salvo et al. [108] performed a field experiment to evaluate the impact of different inoculation and nitrogen fertilization treatments on maize cropping and some specific microbial groups involved in nitrogen and carbon cycles (microaerophilic nitrogen-fixing bacteria and cellulolytic and nitrifying microorganisms) at the vegetative and reproductive stages. For the inoculation treatments, a commercial formulation based on a combination of *A. brasilense* and *Pseudomonas fluorescens* and three experimental formulations based on a single-strain inoculant (*A. brasilense* 40M or *A. brasilense* 42M) or combined inoculants (*A. brasilense* 40M + *A. brasilense* 42M) were used. The work showed that the interaction between the inoculation and fertilization treatments modified the functional diversity of the rhizosphere microbial communities (as revealed by the community-level physiological profile analysis, CLPP) at the reproductive stage. Furthermore, a higher number of nitrogen-fixing rhizobacteria was recorded in plants inoculated with the two strains than those inoculated with *A. brasilense* 40M, suggesting a competitive advantage of the combined inoculants compared to the single-strain inoculant.

Through ARISA fingerprinting analysis, Baudoin et al. [109] observed a transient, but statistically significant, change in the maize rhizobacterial communities upon *A. lipoferum* CRT1 inoculation. A significant impact of the inoculant on the bacterial communities of the samples collected at 7 and 35 days after sowing occurred; multiple changes in the band relative intensity were detected, indicating a broad community shift lasting for at least one month. Florio et al. investigated the possible effects on the microorganisms involved in the nitrogen cycle after *A. lipoferum* CRT1 seed inoculation of maize grown under both conventional and organic farming [110]. Inoculation resulted in a significant denitrifier increase in sites with high carbon limitation, probably due to a stimulation of root carbon exudate release induced by the strain CRT1. On the contrary, the denitrifier abundance slightly decreased in sites with low carbon limitation, likely due to an enhanced competition for nitrate between the roots and denitrifiers.

Recently, Renoud et al. carried out a study to investigate the effect of *A. lipoferum* CRT1 inoculation on maize grown in three sites (characterized by different soil types) in relation to different nitrogen fertilization levels [111] or inoculant levels [112]. The authors investigated the impact on the whole bacterial community and on three notable microbial functional groups (nitrogen fixers, ACC deaminase producers, and 2,4-diacetylphloroglucinol producers). The study revealed that the inoculation with *A. lipoferum* CRT1 promoted maize growth and affected the resident microbiota, despite poor inoculant survival. Inoculation induced changes in the whole bacterial community composition and resulted in a significantly different incidence of the specific bacterial groups (in particular, nitrogen fixers and ACC deaminase producers) at each field site, depending on the cropping year, maize growth stage, nitrogen fertilization, or inoculant levels. Different modifications of the community composition and structure occurred at the three sites, indicating inoculation field-specific effects (related to local abiotic and biotic factors) on indigenous bacteria. Furthermore, it was demonstrated that both standardized inoculant levels and 10-fold reduced levels affected the specific group occurrence and that the inoculant-reduced levels had a greater effect on the resident microbiota (only the reduced formulation significantly impacted the total bacterial community at the three sites).

Coniglio et al. [113] reported that maize seed inoculation with *A. argentinense* Az39 (formerly *A. brasilense* Az39) induced changes in community composition, structure, and functionality. A significant reduction in bacterial community evenness and an increase in the relative abundance of beneficial genera were observed. Beta-diversity analysis showed that the bacterial communities of the inoculated plants clearly differed from those of the non-inoculated plants and bulk soil. In addition, the functional community profiling evidenced an enrichment of chemoheterotrophic and aerobic chemoheterotrophic traits in the communities of inoculated plants compared to non-inoculated plants and bulk soil.

Some field experiments have documented the impact of *Azospirillum* inoculation on rice microbiota. García de Salamone et al. (2010) reported that single-strain inoculation with *A. brasilense* 40 M and *A. brasilense* 42 M (isolated from maize roots) in rice seed resulted in community metabolic profile changes (CLPP analysis) [114]. In two studies using the same host plant cultivar, the impact of *Azospirillum* sp. B510 inoculation on rice microbial communities was investigated through high-resolution molecular methods [115,116]. Bao et al. observed that no marked shifts occurred in the bacterial communities associated with the shoot or base of the rice plants upon inoculation. However, *Azospirillum* sp. B510 inoculation did not affect dominant bacterial groups, but significantly influenced minor bacterial group abundances [115]. Yasuda et al. [116] evaluated the *Azospirillum* sp. B510 inoculation effect at the vegetative and harvesting stages on the bacterial and fungal diversity harbored by the rhizosphere of rice grown under different nitrogen fertilization treatments. *Azospirillum* B510 inoculation did not affect the fungal community structure, but impacted the bacterial communities. Significant changes were observed at the vegetative stage in the bacterial community of rice grown in paddy fields where supplemental nitrogen was not applied—an increase in bacterial diversity, as well as differences in community taxonomic and functional compositions, were shown in inoculated plants compared to the non-inoculated plants. In particular, the inoculation with *Azospirillum* B510 induced an increase in diazotrophic bacteria in the rice rhizosphere under low nitrogen conditions. In addition, inoculation resulted in slight bacterial community shifts at the harvest stage in rice grown in paddy fields where supplemental nitrogen was applied.

Some authors have evaluated the impact of *Azospirillum* inoculants on wheat rhizosphere microbial diversity under greenhouse and field conditions. In open field conditions, wheat treatment with two commercial inoculants containing *Azospirillum brasilense* modified the microbial community’s functional potential at the tillering and grain-filling stages, as evidenced by differences in the carbon substrate utilization patterns (CLPP profiling) [117]. The field experiment performed by di Salvo et al. showed changes in the composition (T-RLFP fingerprinting) and physiological profiles (CLPP analysis) of the wheat rhizobacterial community upon single-strain inoculation with *A. brasilense* 40M and 42M at the jointing stage, but not at the grain-filling stage [118]. Baudoin et al. [119] assessed the effects of wildtype and genetically modified inoculants on wheat rhizosphere microbiota through greenhouse experiments. The IAA overproductive *A. baldaniorum* Sp245 (formerly *A. brasilense*) mutants contained construct with the ipdC gene (encoding an indole-3-pyruvate/phenylpyruvate decarboxylase) under the control of a constitutive or a root exudate-inducible promoter. ARISA fingerprinting analysis showed that no statistically significant differences were observed among the bacterial communities of non-inoculated and inoculated (either with wild-type or mutant strains) plants. However, significant differences were found between the bacterial communities of wheat inoculated with the mutant with the ipdC gene under the control of the inducible promoter and the wheat inoculated with the wild-type strains. Instead, differences were found between the fungal communities of wheat inoculated with both mutants and the wheat inoculated with the wild-type. In addition, inoculation with the two mutants resulted in similar effects on the native fungal biota.

Two studies displayed the variable effects of single-strain inoculation and/or co-inoculation on the rhizosphere microbiota of tomatoes (experiments were performed in growth chambers). Correa et al. [120] documented the different effects of *A. brasilense* BNM65 inoculation on the rhizoplane bacterial community of two different tomato varieties at 60 days after planting. Inoculation modified the carbon source utilization profile of the bacterial communities associated with the rhizoplane of both cherry and fresh-market tomatoes. The indigenous rhizoplane bacterial communities of non-inoculated cherry and fresh-market tomatoes showed similar DGGE banding patterns. While the inoculation induced shifts in the bacterial community of cherry tomatoes, no changes were evidenced in those of fresh-market tomatoes. Felici et al. [121] investigated the impact on the tomato rhizosphere microbial community at 45 days after sowing of individual and combined applications of *A. baldaniorum* Sp245 and *Bacillus subtilis* 101. DGGE analysis showed that the bacterial and fungal community patterns of the plants treated with the co-inoculant formulation were very similar to those of the non-inoculated tomatoes. Instead, the single-strain inoculation seemed to have only marginal effects on the dominant bacterial and fungal populations. In addition, the sequencing of the recovered bands revealed that inoculation did not interfere with other root-colonizing bacteria (e.g., *Bradyrhizobium* and *Sphingomonas* species).

*Azospirillum* inoculation effects on the resident microbiota were also investigated in intercropping systems. Pardo-Diaz et al. (2021) evaluated the effect of single-strain inoculation and co-inoculation with *Azospirillum brasilense* D7 and *Herbaspirillum* sp. AP21 in a grass–legume intercropping system under reduced nitrogen fertilization levels (greenhouse experiments) [122]. The metabarcoding profiling of the rhizosphere bacterial diversity evidenced that the inoculation altered the bacterial community composition and structure. A significant alpha-diversity reduction was observed in the crops inoculated with the strain D7, whereas no significant changes were observed upon co-inoculation. Furthermore, it was evidenced that the rhizosphere bacterial community shift induced by biofertilizer inoculation correlated with an improvement in crop growth and quality (crude protein, shoot nitrogen content, and shoot dry weight increases).

### 3.2. Effects of Azotobacter

*Azotobacter* spp. is a genus including Gram-negative bacteria belonging to *Gammaproteobacteria* and commonly found in the soil and rhizosphere of different crop plants [123]. This genus includes free-living aerobic bacteria able to fix atmospheric nitrogen and solubilize phosphate, and then induce plant-beneficial effects, mediated by the production and release of active compounds such as hormones, vitamins, siderophores, and amino acids [124]. Due to their ability to fix molecular nitrogen and stimulate plant growth, *Azotobacter* species are widely used in agriculture as biofertilizers. In fact, the plant inoculation with *Azotobacter*, in consortia with other bacterial species or used as a single inoculant, favored seed germination, growth, and proliferation of different crop plants [125].

Moreover, strains belonging to the *Azotobacter* genus are known for their ability to degrade pesticides and the derivative of aromatic compounds and for tolerating high concentrations of salt. In fact, different studies have demonstrated the ability of some *Azotobacter* strains to promote and improve the growth of crop plants, such as maize [126] and wheat [127] under salt stress conditions.

Currently, many papers report the effect of this bacterium on the growth promotion and yield increase of various plants. However, few studies can be found in the literature regarding the impact of *Azotobacter* strains, inoculated alone or in consortium, on the indigenous bacterial communities of soil and associated with plants.

One of the rare papers on this topic was published recently by Sharma and collaborators [128] who studied the impact of a bacterial consortium composed of *Azotobacter chrooccum*, *Bacillus megaterium*, and *Pseudomonas fluorescens* on the nutrient uptake of *Cajanun cajan* and its possible non-target effects on the composition and active fraction of native bacterial community by DGGE and 16S rRNA transcripts (qPCR) analysis, respectively. The main result of this paper was that the resident bacterial community remained unaffected by plant inoculation with the consortium. However, the authors speculated that the low sensitivity of the DGGE technique overshadowed the possible modifications that occurred in the bacteriome. On the contrary, the abundance of the active fraction of the bacterial community was modulated by the plant treatment, with an increase of *nifH* gene transcripts during the phenological stages corresponding to flowering and maturity as the main effects observed [128]. These data clearly indicated that plant inoculation with *A. chrooccum, B. megaterium*, and *P. fluorescens* did not modify the whole structure of the microbial community, but stimulated nitrogenase expression by active plant-beneficial bacteria, thus improving the nitrogen turnover in soil.

More recently, the impact of the same bacterial consortium on the abundance and structure of microbial communities in the rhizosphere of tall fescue and pigeon pea and bulk soil was investigated in the presence or absence of the human pathogen *Listeria monocytogenes*. The results obtained in this study demonstrated a shift in the resident soil bacterial communities, with a significant inhibition of the *L. monocytogenes* cell density, a reduction of taxa corresponding to phytopathogen, and a correspondent increase of bacterial species known as biocontrol agents and PGPB [129].

### 3.3. Effects of Rhizobium *spp*.

Rhizobia are Gram-negative nitrogen-fixing bacteria belonging to the *Alphaproteobacteria* that can live in soil as free-living PGPB or associated with legumes (such as soybean, chickpea, lentil, pea, common bean, alfalfa, and clover) as parts of the rhizobia–legume symbioses. As saprophytes, rhizobia survive in a complex microbial community by adopting an oligotrophic lifestyle, while inside the host legume, they differentiate into endosymbiotic bacteroids [130].

Much of the research regarding rhizobia has focused on the specific structure or mechanisms of the rhizobia–legume symbiosis, without going deeper to the role of rhizobia as members of complex soil communities. In fact, rhizobia survive in soil and compete with the local microbiota before establishing the symbiosis. The literature is lacking a clear spatial and temporal evolution of the various stages in the genetic and metabolic modulation during the pre-symbiotic stages. Most studies focused their attention more on the effects induced by a single rhizobia strain on plant growth and development than on the relationships occurring with the rhizosphere microbial communities. So, Trabelsi and collaborators [131] reported that when using nitrogen-fixing bacteria to increase productivity in field conditions, some problems should arise during symbiosis establishment. In some cases, in fact, the rhizobia population living in the soil tends to compete with the resident microbiota. Therefore, in order to guarantee a proficient symbiosis establishment, the inoculated bacteria must, above all, be a good competitor. In this regard, some molecular analyses clearly demonstrated that the efficacy of rhizobia inoculation is correlated with the abundance of *nifH* genes (encoding nitrogenase reductase), typically occurring in the late flowering period of alfalfa [132]. Other genes involved in nitrogen turnover are affected by inoculation; for example, the high number of copies of the *amoA* (ammonia monooxygenase) gene are reported during alfalfa flowering when an effective nitrogen-fixing strain has been inoculated [131,132].

So, what happens to the microbial communities of the rhizosphere when a high level of one or more rhizobia strains is introduced into the rhizosphere of leguminous plants? The answer is “it depends”. For example, Trabelsi and coworkers (2011) compared inoculated and uninoculated plants of *Phaseolus vulgaris* and observed that co-inoculation (*E. meliloti* 4H41 and *R. gallicum* 8a3) provides an excellent result in stimulating microbial communities. In fact, an increase in the ribotypes attributable to the inoculated strains is observed [131]. However, when studying the impact on resident microbial communities, it must be kept in mind that other perturbing factors besides inoculation, such as soil type, cultivated species, and anthropogenic effects such as agricultural practices, can behave as disturbing variables in the system.

An interesting review by Trabelsi and Mahmdi [133] summarizes the impact of rhizobia-based inoculants on soil microbial communities. In particular, it is reported that Schwieger and Tebbe [134] observed how the inoculation with the *Sinorhizobium meliloti* L33 strain influences the biodiversity in the rhizosphere of *Medicago sativa* both qualitatively and quantitatively, inducing an increase in *Alphaproteobacteria* and a reduction in *Gammaproteobacteria* [135]. Hermann et al. observed the stimulating effect of a cocktail of *Ensifer* sp. strains (belonging to the *Sinorhizobium* group) in the field on the total bacterial diversity in the *Acacia senegal* rhizosphere; however, this increase was also related to seasonal variations. This ambiguous result is probably attributable to the low performance of the DGGE technique [135]. Moreover, it is also reported that in the case of *S. meliloti* M401/M403 inoculation, the persistence of certain γ-proteobacterial populations in the rhizosphere of alfalfa could be affected using a TGGE approach [136]. Using a T-RFLP approach, significant effects on bacterial structure and diversity in the bulk soil of common bean were observed after plant inoculation with *Rhizobium gallicum* 8a3 and *Ensifer meliloti* 4H41 in field conditions. The abundance of both 𝛼- and γ-proteobacteria, together with *Firmicutes* and *Actinobacteria*, was increased after dual inoculation. Similarly, the density of plant-beneficial microorganisms such as *Rahnella, Bacillus, Azospirillum, Mesorhizobium, Pseudomonas, Streptomyces*, and *Sinorhizobium* was higher in plants inoculated with strains 8a3and/or4H41 than in untreated plants subjected to nitrogen-fertilized treatment [131,133]. The results obtained indicated that the effect induced by plant inoculation with PGPB on resident bacterial communities is more intense and more durable than those induced by nitrogen chemical fertilization. In recent work carried out using the NGS approach, it has been observed that in the soybean rhizosphere, the rhizobial community composition in nodules may also be influenced by some other members of the rhizosphere microbiota, such as *Bacillaceae*, which may be involved in the colonization of nodules by *Sinorhizobium* and *Bradyrhizobium* [137]. Finally, Checcucci and Marchetti [138] clearly resumed the inter-kingdom and intra-kingdom communication involving plants and microbes in the rhizosphere, underlying the consistent role of rhizobia. In addition, the authors reported that recent data collected under identical soil conditions demonstrated that the plant genotype, through its phenotypic features, can filter and modulate the microbial community structure and function, as well as the diversity of root-associated bacteria [139,140] carrying out a partner choice in rhizobia mutualism [141].

### 3.4. Effects of Other Bacterial Species

Due to their role in plant growth promotion and their features of biological control agents, the genera *Pseudomonas* and *Bacillus* are well known for their possible application as biofertilizers.

As previously reported, the literature is not rich with papers dealing with the possible non-target effects of bacterial plant inoculants on the community structure. However, some work focused attention on this topic. An analysis of the literature starting from the work of [142], who analyzed the effect of the strain *P. fluorescens* 2P24, isolated from a suppressive soil against take-all, on the composition of rhizospheric microbiota using DGGE and T-RFLP techniques, was reported. The inoculation of this bacterial strain transiently affects the fungal community of the cucumber rhizosphere: while just after inoculation, a strong effect on the fungal community was detected, the intensity of this impact tended to reduce as the concentration of the strain P24 decreased [142]. Moreover, both *P. fluorescens* 2P24 and *P. fluorescens* CPF10 led to the temporary alteration of the bacterial community associated with cucumber root. More in detail, shifts in microbial community composition were observed from the 7th to 42nd day after treatment with strains 2P24 or CPF10 for *Cyanobacterium*, *Betaproteobacterium*, *Staphylococcus*, and *Bacillus*. However, these modifications disappeared on the 56th day after inoculation [143]. Similar transient effects were previously observed on the rhizosphere microbiome of *Hordeum vulgare* (barley) after 3 weeks post-inoculation with the *Pseudomonas* sp. DSMZ 13,134 strain [144].

The effect of the biocontrol strain *B. amyloliquefaciens* FZB42, active against the phytopathogenic fungus *Rhizoctonia solani* and able to synthesize cyclic lipopeptides (CLP) and polyketides, on the native bacterial communities of lettuce was studied in two studies performed by terminal restriction fragment length polymorphism (T-RFLP) [145] and by the high-throughput metagenome sequencing of whole-community DNA [146]. The main result of these papers is that, although a general shift of the rhizosphere microbiota was observed according to plant age, lettuce inoculation with FZB42 did not show any long-term significant effect on the resident rhizosphere bacterial community.

After a first evaluation of the effects induced by 54 *Pseudomonas* strains on lettuce plants grown under field conditions, Cipriano et al. [147] identified as PGPB two isolates, IAC-RBru1 and strain IAC-RBcr4, isolated from rucola and chrysanthemum, respectively. Then, the authors decided to assess the possible shifts in the microbial community structure due to the plant inoculation of the two strains on lettuce. The main results highlighted the fact that the introduction of the two strains induced an increase of the abundance of *Verrucomicrobia*, candidate division BRC1, *Armatimonadetes*, and *Nitrospirae*. Moreover, in the rhizosphere of plants inoculated with the strain IAC-RBru1, a higher amount of Planctomycetes was also observed [147]. At the genus level, *Pirellula*, *Fonticella*, *Anaerolinea*, and *Isosphaera* increased in the rhizospheres of plants treated with strain IAC-RBcr4, while the isolate IAC-RBru1 enhanced the abundance of *Anaerolinea*, *Isosphaera*, *Singulisphaera*, *Lentibacillus*, *Kribella*, and *Siphonobacter*.

One year later, Qin et al. [148] showed a modification of the composition of the microbial communities associated with cucumber roots after inoculation with *Bacillus amyloliquefaciens* L-S60, especially concerning specific bacterial taxa such as *Bacillus*, *Rhodanobacter*, *Paenibacillus*, *Pseudomonas*, *Nonomuraea*, and *Agrobacterium*, which were increased in their concentration.

Plant inoculation with PGPB can affect not only the rhizosphere community, but also the populations living inside plant tissues. The impact on the diversity and richness of the endophytic microbial communities in broccoli was characterized through a metagenomic approach after inoculation with *B. cereus, B. subtilis*, and *B. amyloliquefaciens* alone or in a consortium [149]. Interestingly, while the Bacillus strains did not enter the plant tissue, they induced changes in the abundance of some bacterial groups. As an example, *Pseudomonadales*, *Rhizobiales*, *Xanthomonadales*, and *Burkholderiales* were dominant in the broccoli endosphere, but inoculation with *B. amyloliquefaciens* reduced the frequency of *Pseudomonas* and increased the density of several of minor taxa.

Ke et al. [150] analyzed the effect of *P. stutzeri* A1501 and *nifH* mutant 1502 on the plant growth and microbial communities of maize under two different water regimes through MiSeq sequencing and qPCR of functional genes and transcripts (*nifH* and *amoA*) involved in the nitrogen cycle. Besides promoting the growth of the plants and enhancing the nitrogen accumulation level, plant treatment with *P. stutzeri* A1501 significantly affected the composition of the diazotrophic community and induced an increase of resident diazotrophs and ammonia oxidizers and their respective functional gene transcripts [150].

More recently, the effect of *Bacillus subtilis* and *Pseudomonas fluorescens*, as single and co-inoculation, was investigated on lettuce cultivated under organic and conventional managements. The results obtained showed an increased biomass of bacteria, both Gram-negative and -positive, fungi, and *Actinomycetes* in the organic soil treated with the strains, suggesting a lack of competition between the two bacterial inoculants and the resident microflora, possibly related to an enhanced availability of nutrients through rhizodeposition [151].

The impact of the inoculation of *P. fluorescens* LBUM677 on the rhizosphere microbiome of *Brassica napus*, *Buglossoides arvensis*, and *Glycine max* after 30-, 60- and 90-days post-inoculation was analyzed by Illumina sequencing. The results showed that both the alpha and beta diversity of the rhizosphere microbiome of the three oilseed crops changed according to the time post inoculation and to the plant treatment with the strain LBUM677. Comparing the taxa differential abundance, the authors highlighted the fact that 29 bacterial groups *(Actinobacteria Alphaproteobacteria, Gammaproteobacteria, Bdellovibrionota, Armatimonadota*, and *Bacteropodota*) were more abundant in the rhizosphere of inoculated plants, while the frequency of another 30 taxa (*Acidobacteria, Chloroflexi, Planctomycetota, Elusimicrobiota, Methylomirabilota, Fibrobacterota*, and *Cyanobacteria*) was higher in the untreated plants [152].

Very recently, the impact of the strain *Bacillus subtilis* T6-1 on the structure of the bacterial and fungal communities associated with the poplar rhizosphere was investigated by Illumina MiSeq sequencing [153]. After seedling inoculation, an increase of the culturable bacterial fraction density and a reduction of the fungal ones, coupled to a reduction in fungal biodiversity, occurred. Overall, the data did not highlight significant modifications in the composition of the bacteriome following T6-1 inoculation.

### 3.5. Effects of Bacterial Consortia

The effect of the inoculation of bacterial consortia on the growth and increase of plant biomass has been studied for years. The idea of using bacterial consortia is based on the possibility of exploiting bacterial strains, compatible with each other, with diverse plant-beneficial traits. Once more, the number of papers dealing with the non-target effect of plant inoculation with bacterial consortia on resident microflora is quite low. In this section, we analyze some of this work.

To the best of our knowledge, the first work that reported the impact of bacterial consortia on microbial communities was conducted by Andreote et al. [154]. The authors analyzed the effect of two spontaneous rifampicin endophytes, *Paenibacillus* sp. E119 and *Methylobacterium mesophilicum* SR1.6/6, on three cultivars of potato. The results analyzed by multivariate statistical analysis demonstrated that the structure of the bacterial communities was mainly affected by the potato cultivar and the plant developmental stage. On the other hand, the *Alphaproteobacteria* and *Paenibacillus* taxa changed according to the bacterial inoculation with the consortium [154].

In a second paper [155], *Chamomilla recutia* seedlings were inoculated with two bacterial consortia, one composed of three indigenous Gram-positive strains (*Streptomyces subrutilus* Wbn2-11, *Bacillus subtilis* Co1-6, and *Paenibacillus polymyxa* Mc5Re-14) and one based on three Gram-negative strains (*Pseudomonas fluorescens* L13-6-12, *Stenotrophomonas rhizophila* P69, and *Serratia plymuthica* 3Re4-18). After 4 and 8 weeks of growth, the rhizosphere samples were analyzed by pyrosequencing and the results obtained showed an increase of *Acidobacteria* in the samples treated with the Gram-positive consortium compared to the uninoculated controls. Moreover, a different abundance in the composition of the genera *Rhizobium*, *Pseudomonas*, *Flavobacterium*, and *Arthrobacter* was detected according to the bacterial inoculum [155].

In the work by Thokchom et al. [156], the effect of the combined and single inoculation of *Enterobacter hormaechei* RCE1, *Enterobacter asburiae* RCE2, *Enterobacter ludwigii* RCE5, and *Klebsiella pneuomoniae* RCE7 on the growth and the associated resident epiphytic and endophytic bacterial community associated with mandarin orange seedling roots was studied after 90, 180, and 360 days. The PCR-DGGE analysis showed that PGPB inoculation, and, to a lesser extent, plant age, affected the resident bacterial communities in the rhizosphere and within the plant tissues as well [156]. Unfortunately, the limits imposed by the technique itself did not allow for a better understanding of the shifts induced on the resident bacteriome.

One year later, Wang et al. [157] analyzed the effect of a bacterial consortium based on *Ensifer* sp. NYM3, *Acinetobacter* sp. P16, and *Flavobacterium* sp. KYM3 on cucumber plants. The authors produced two consortia using different ratios of the same bacterial strains according to cucumber nutritional requirements and soil fertility levels. The impact of these consortia on the resident microbial communities was analyzed by amplification of the V4 hypervariable region followed by Illumina sequencing. Besides the growth stimulation, plant inoculation with the consortia resulted in an increase of *Gammaproteobacteria*, *Nitrospirae*, and *Acidobacteria* and a reduction in *Actinobacteria* and *Firmicutes* compared to the control [157].

In 2019, Zhang et al. [158] investigated the impact of a consortium of three bacterial strains, PGPB and biocontrol agents (*Bacillus cereus* AR156, *Bacillus subtilis* SM21, and *Serratia* sp. XY21) on sweet pepper plants infected by *Phytophora capisci*. The results obtained by the authors showed that the application of the inoculum, in addition to reducing the disease, induced shifts in the resident bacteriome. In particular, the results obtained by Illumina sequencing showed a significant increase of the genera *Burkholderia*, *Comamonas*, and *Ramlibacter* compared to the control [158].

Finally, the impact of *Bacillus cereus* BT-23, *Lysobacter antiobioticus* 13-6, and *Lysobacter capsici* ZST1-2 inoculated alone or in combination in Chinese cabbage plants infested with *Plasmodiophora brassicae* was evaluated after 60 days. The disease was contained by 66% and at the same time, according to the results obtained by sequencing, the structure of the bacterial community was modulated according to the different treatments. Altogether, these data indicated that the inoculation with the microbial consortium reduce the disease rate through the recovery of the imbalance occurring within the microbial community [159].

## 4. Conclusions and Perspectives

Considering all of the literature analyzed in this review, the possible non-target effects induced by PGPB used as biofertilizers or biocontrol agents are usually transient and directed towards specific bacterial taxa. On the other hand, it is also true that soil management (through fertilization, compost or biochar introduction, or the application of fungicide) is not without side effects on the microbial community.

As stated several times during this review, the literature regarding this topic is limited mainly due to the past low availability of efficient tools to study biodiversity. Of course, the development of -omics techniques allowed scientists to open the “black box” containing the soil biodiversity; this is providing a huge amount of information regarding the interactions between possible new plant bioinoculants and the soil/rhizosphere microbiome. Another stimulating idea is that PGPB that are able to synthesize volatile and quorum-sensing molecules involved in plant growth promotion and in stress tolerance can modulate the interactions among bacteria, as well as between bacteria and fungi or with the host plant, thus affecting the structure and dynamics of the resident bacteriome. These topics have been less explored [160,161] and deserve more attention from the scientific community. Moreover, based on the idea that an efficient root colonization is a prerequisite for the success of PGPB, another critical issue to be addressed is the development of new strategies to detect and quantify the PGPB once inoculated in soil, on seeds, or on the plant. Following the implementation of these results, it will be easier to understand the real weight of these non-target effects. Surely, more efforts must be given toward developing new biofertilizers based on PGPB belonging to other species besides those already accepted by the EU regulations. We firmly believe that this is vital, especially considering the already evident environmental stresses induced by global climate changes.

## Figures and Tables

**Figure 1 microorganisms-10-02462-f001:**
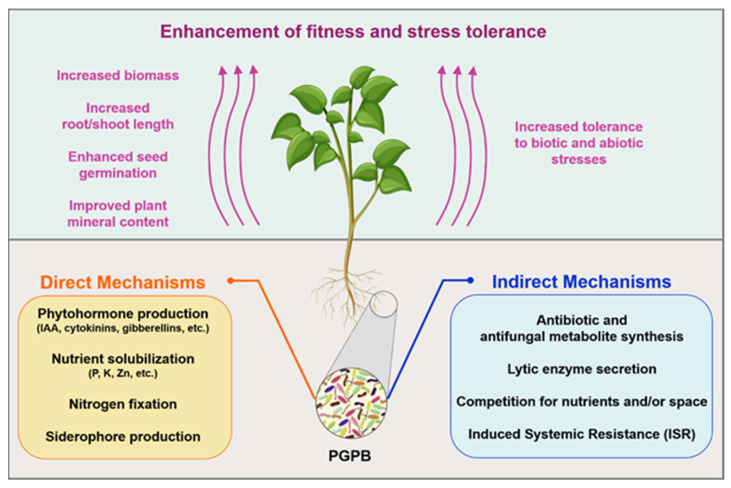
Overview of the main mechanisms used by PGPB to improve plant growth.

**Figure 2 microorganisms-10-02462-f002:**
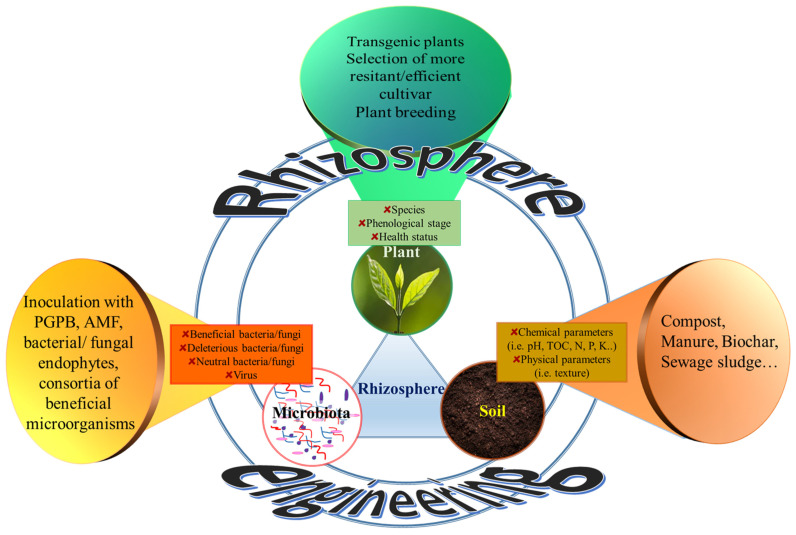
Graphical representation of rhizosphere engineering. At the center of the concept is the rhizosphere trinity involving the soil, the plant, and the associated microbiota and the relations occurring among these three components. The rhizosphere environment, the plant development, and the structure of the associated microbial community depend on plant and soil parameters, as well as on the microbiota composition. The ways to engineer the rhizosphere involve the manipulation of the microbiota, the genetic plant modifications and selection, and the modification of soil parameters. Some examples of the tools available to manipulate the three compartments are reported in the cones.

**Figure 3 microorganisms-10-02462-f003:**
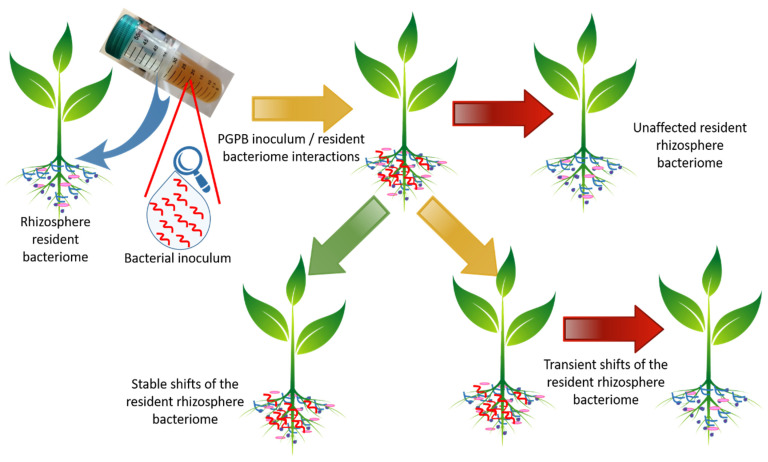
Graphical representation of the possible outcomes of plant inoculation with PGPB. Following plant inoculation, the interaction with the plant and the associated microflora can result in stable or transient non-target effects on the resident bacteriome. If the impact is temporary, in a relatively short time, the bacteriome restores its initial condition. Alternatively, the resident bacteriome can be unaffected by plant inoculation, so that non-target effects do not occur.

## Data Availability

Not applicable.

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
