# Peer review of "Impact of Plant-Beneficial Bacterial Inocula on the Resident Bacteriome: Current Knowledge and Future Perspectives"

_microorganisms, 2022, doi:10.3390/microorganisms10122462_

Round 1

Reviewer 1 Report

In this review the authors discuss the effect of PGPB addition on native bacterial communities focusing on existing rules. It is an interesting and original topic to take under study and it indeed addresses the main question posed. As a review manuscript it provides an analysis of the current situation. The conclusions are consistent with the evidence and arguments presented.  All the references are appropriate.

Author Response

Reviewer 1

In this review the authors discuss the effect of PGPB addition on native bacterial communities focusing on existing rules. It is an interesting and original topic to take under study and it indeed addresses the main question posed. As a review manuscript it provides an analysis of the current situation. The conclusions are consistent with the evidence and arguments presented.  All the references are appropriate.

A: We thank the reviewer for taking the time to critically read the manuscript and for his appreciation of our work

Reviewer 2 Report

The manuscript presents an extensive presentation of reports about the effect of inoculations of PGPB on plant development with the emphasis on posiible effects on the accompanying bacterial rhizosphere community. It became obvious that there are some contradictions in the findings (influence on bacteriome found vs. no influence) which may in part be due to different methods applied, which enable different sharpness and depth in identifying changes in the resident bacteriome.  Recently, also first results on the effects of PGPB-inoculations on fungal communities were published, but there are just very few reports and a comparative survey should be the subject of later studies. 

There should be an important information added in the  1. Introduction about methods of strain-specific identification of bacterial inoculum in a very diverse microbial background, which is a prerequisite for the validation of the success of an inoculation - and posiible effects of the resident bacteriome. For example, Stets et al. (Appl. Env. Microbiol. 81, 6700-6709, 201) described a successful quantification of Azospirillum brasilense strain FP2 in wheat based on genome-based strain-specific qPCR.

Part 2 is missing.

Furthermore, in the part 3 "PGPR mechanisms", also presented in Figure 1, the description of the plant growth promotion mechanisms should be extended. Due to stimulatory effects on root development (mostly by hormones), improved water uptake was found very important in soils with limited water supply. Concomittantly, the uptake of soluble nutrients from soil, like potassium, nitrate etc. was found improved as result from the inoculation effect (direct mechanism). Furthermore, the influence of bacterial metabolites, like volatiles (B.H.S. Dias et al., C4-bacterial volatiles improve plant health, Pathogens 2021, 10,682) and quorum sensing signals like N-acylhomoserine lactones (A. Schikora et al., Beneficial effects of bacteria-plant communication based on quorum sensing molecules. Plant Mol. Biol. 90, 605-612, 2016) results in improved plant development and abiotic and biotic stress tolerance (indirect mechanism). Inoculation with bacteria harbouring these interactive molecules may also lead in influences on the resident bacteriome, since it influences biofim and secondary  metabolite synthesis This could be mentioned in the conclusion for future studies. 

Author Response

The manuscript presents an extensive presentation of reports about the effect of inoculations of PGPB on plant development with the emphasis on posiible effects on the accompanying bacterial rhizosphere community. It became obvious that there are some contradictions in the findings (influence on bacteriome found vs. no influence) which may in part be due to different methods applied, which enable different sharpness and depth in identifying changes in the resident bacteriome.  Recently, also first results on the effects of PGPB-inoculations on fungal communities were published, but there are just very few reports and a comparative survey should be the subject of later studies. 

There should be an important information added in the  1. Introduction about methods of strain-specific identification of bacterial inoculum in a very diverse microbial background, which is a prerequisite for the validation of the success of an inoculation - and posiible effects of the resident bacteriome. For example, Stets et al. (Appl. Env. Microbiol. 81, 6700-6709, 201) described a successful quantification of Azospirillum brasilense strain FP2 in wheat based on genome-based strain-specific qPCR.

A: We added this interesting observation in the conclusion section.

Part 2 is missing.

A: We thank Reviewer 1: we corrected the typo.

Furthermore, in the part 3 "PGPR mechanisms", also presented in Figure 1, the description of the plant growth promotion mechanisms should be extended.

A: We don’t agree with the reviewer. The mechanisms used by PGPB have been already well described in the literature and the amount of papers dealing with this topic is really impressive. In this section our idea was to provide a brief overview of the mechanisms at the base of plant growth promotion and use this paragraph as a link to the other sections which are the real target of the review.

Due to stimulatory effects on root development (mostly by hormones), improved water uptake was found very important in soils with limited water supply. Concomittantly, the uptake of soluble nutrients from soil, like potassium, nitrate etc. was found improved as result from the inoculation effect (direct mechanism). Furthermore, the influence of bacterial metabolites, like volatiles (B.H.S. Dias et al., C4-bacterial volatiles improve plant health, Pathogens 2021, 10,682) and quorum sensing signals like N-acylhomoserine lactones (A. Schikora et al., Beneficial effects of bacteria-plant communication based on quorum sensing molecules. Plant Mol. Biol. 90, 605-612, 2016) results in improved plant development and abiotic and biotic stress tolerance (indirect mechanism). Inoculation with bacteria harbouring these interactive molecules may also lead in influences on the resident bacteriome, since it influences biofim and secondary  metabolite synthesis This could be mentioned in the conclusion for future studies. 

A: We thank the reviewer for the suggestion: we have added this exciting idea in the conclusions section and the two suggested papers in the references.

Reviewer 3 Report

I consider this document is very well written, it presents an extensive and sufficient state of the art that includes the possible impact of the PGPB deliberate release on the resident microbiota, and the main effects of the PGPB in plants, with the purpose of continuing to develop products to be applied in agriculture. In the PDF document, I include small suggestions to improve the document. In my opinion, the manuscript is suitable for publication.

Author Response

Reviewer 3

I consider this document is very well written, it presents an extensive and sufficient state of the art that includes the possible impact of the PGPB deliberate release on the resident microbiota, and the main effects of the PGPB in plants, with the purpose of continuing to develop products to be applied in agriculture. In the PDF document, I include small suggestions to improve the document. In my opinion, the manuscript is suitable for publication.

A: We thank the reviewer for taking the time to critically read the manuscript and for his appreciation of our work. All the suggestions have been taken into consideration and the manuscript has been modified accordingly
